# CroCo: Self-Supervised Pre-training for 3D Vision Tasks by Cross-View Completion

**Philippe Weinzaepfel**  **Vincent Leroy**  **Thomas Lucas**

**Romain Brégier**  **Yohann Cabon**  **Vaibhav Arora**  **Leonid Antsfeld**

**Boris Chidlovskii**  **Gabriela Csurka**  **Jérôme Revaud**

NAVER LABS Europe
https://europe.naverlabs.com/research/computer-vision/croco/

## Abstract

Masked Image Modeling (MIM) has recently been established as a potent pre-training paradigm. A pretext task is constructed by masking patches in an input image, and this masked content is then predicted by a neural network using visible patches as sole input. This pre-training leads to state-of-the-art performance when finetuned for high-level semantic tasks, *e.g.* image classification and object detection. In this paper we instead seek to learn representations that transfer well to a wide variety of 3D vision and lower-level geometric downstream tasks, such as depth prediction or optical flow estimation. Inspired by MIM, we propose an unsupervised representation learning task trained from *pairs* of images showing the same scene from different viewpoints. More precisely, we propose the pretext task of *cross-view completion* where the first input image is partially masked, and this masked content has to be reconstructed from the visible content and the second image. In single-view MIM, the masked content often cannot be inferred precisely from the visible portion only, so the model learns to act as a prior influenced by high-level semantics. In contrast, this ambiguity can be resolved with cross-view completion from the second unmasked image, on the condition that the model is able to understand the spatial relationship between the two images. Our experiments show that our pretext task leads to significantly improved performance for monocular 3D vision downstream tasks such as depth estimation. In addition, our model can be directly applied to binocular downstream tasks like optical flow or relative camera pose estimation, for which we obtain competitive results without bells and whistles, *i.e.*, using a generic architecture without any task-specific design.

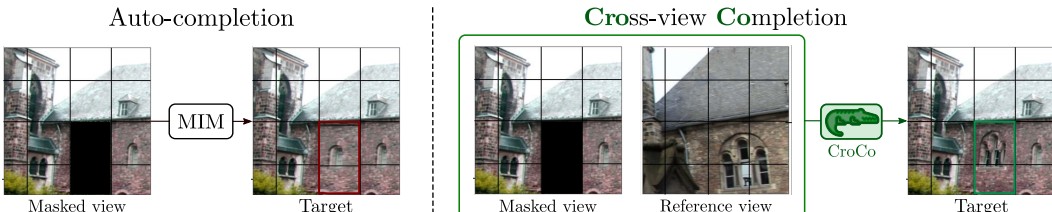

Figure 1: **Auto-completion *vs*. Cross-view completion tasks.** *Left:* given a masked image, a model trained for auto-completion can only leverage the visible context to fill-in the blanks and thus relies mainly on high-level semantic information. *Right:* given an additional view of the same scene, cross-view completion makes precise reconstruction possible, assuming that the model is able to understand both the scene geometry and the spatial relationship between the two images.

36th Conference on Neural Information Processing Systems (NeurIPS 2022).

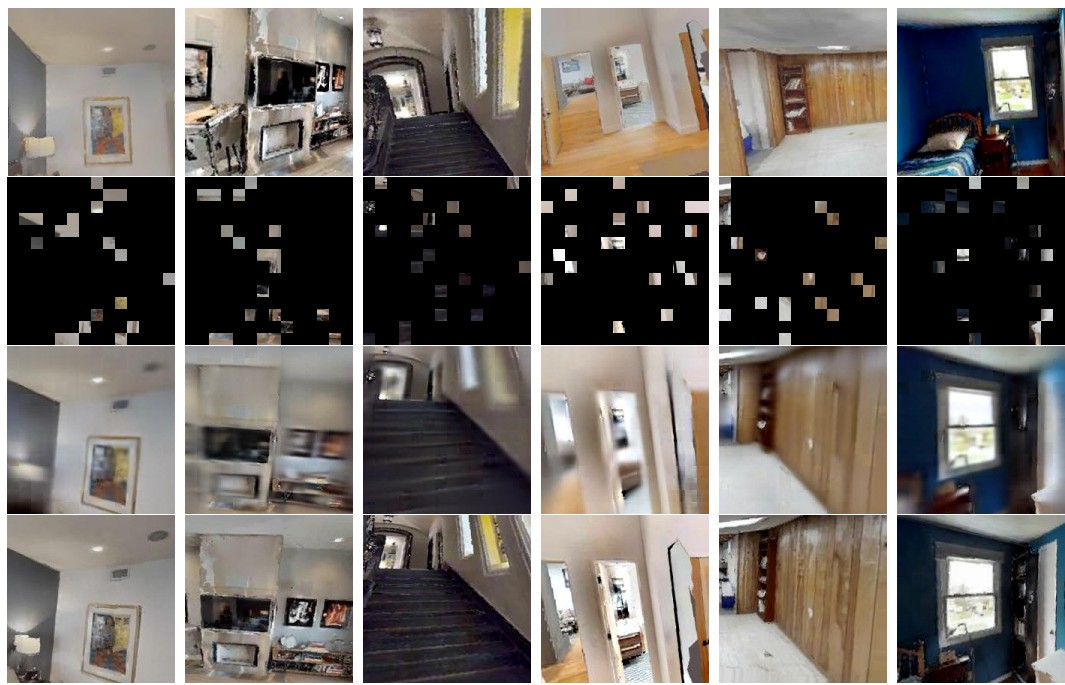

Figure 2: **Reconstruction examples from CroCo** on scenes unseen during training. From top to bottom: reference image (input), masked image (input), CroCo output, original target image.

# 1 Introduction

Self-supervised learning for model pre-training has allowed to achieve state-of-the-art performance in a number of high-level computer vision tasks, such as image classification or object detection. Contrary to traditional supervised learning, these models enable the use of unlabelled data via carefully designed pretext tasks. A popular method for self-supervision is instance discrimination [10, 11, 14, 23, 31, 33, 68] which constructs a pretext task by learning representations that are invariant to various data augmentations. More recently, Masked Image Modeling (MIM) [7, 13, 16, 27, 32, 49, 77] has emerged as a powerful alternative for self-supervision. Inspired by BERT [19], these models are trained using an auto-completion pretext task. An encoder, usually a Vision Transformer (ViT) [21], takes a partial view of an image input, obtained by splitting the image into patches and masking some of them, and encodes it into a latent representation. The masked patches are then predicted by a decoder using the latent representation. To solve the task, the model must leverage the context given by the visible portion and act as a prior for the ambiguous content that cannot be deduced from them. In practice these models are typically trained on object-centric datasets such as ImageNet [55] and thus tend to learn high-level semantic information; that makes them well suited for tasks such as image classification or object detection [4, 32, 37].

In this paper, we propose a self-supervised training objective specially designed to learn 3D geometry from unlabeled data, named *Cross-view Completion*, or *CroCo* in short. Given two images depicting the same scene, random parts of the first input image are masked and then predicted by the model using both (1) the visible parts of this first image, as well as (2) a second image called *reference* image as it depicts the same scene from a different point of view, see Figure 1. While multi-view image completion has a long history in image editing [17, 76], we are the first to explore its potential as a self-supervised representation learning tool. In contrast to single-view completion, the proposed pretext task of cross-view completion allows to perform masked image modeling conditionally on a second view. In this case, most of the ambiguity can be resolved by reasoning about the scene geometry and spatial relationship between the two views. This is illustrated in the reconstruction examples of our model in Figure 2.

Figure 3 provides an overview of our self-supervised model during pre-training. We divide both images into sets of non-overlapping patches, denoted as tokens. Most tokens from the first image are randomly discarded, and the remaining ones are fed to an image encoder, which we implement

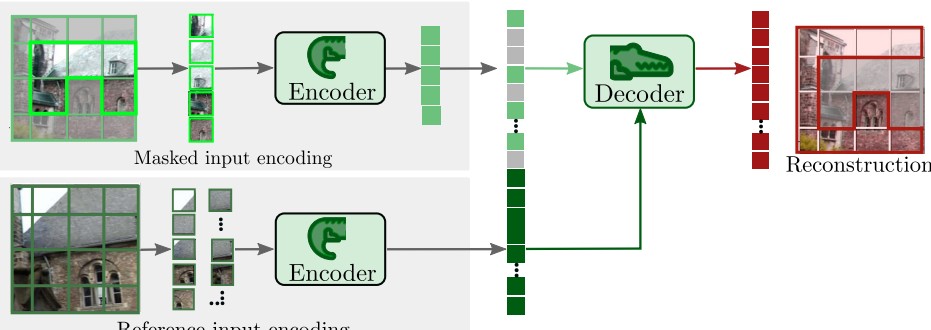

Figure 3: **Overview of our CroCo model during pre-training.** Patches from the first input image are partially masked; visible ones are encoded into their latent representations, then padded with masked tokens to account for hidden patches. The same encoder is used to encode the patches of the second image. The decoder receives the encoded tokens from both images and use them to reconstruct the masked parts of the first image.

using a Vision Transformer (ViT) backbone [21]. All patches from the second (reference) image are encoded in a *Siamese* manner, *i.e.*, using the same encoder with shared weights. The token latent representations output by the encoder from both images, including tokens to account for the masked patches of the first image, are then fed to a decoder whose goal is to predict the appearance of hidden patches. To this aim, we use a series of transformer decoder blocks comprising cross-attention layers. This allows non-masked tokens from the first image to attend tokens from the reference image, thus enabling cross-view comparison and reasoning. Our model is trained using a simple pixel reconstruction loss over all masked patches, similar to MAE [32]. To finetune the model on downstream tasks that process only a single image, *e.g.* monocular depth estimation, the decoder can be discarded and we use the pre-trained encoder alone. In the case of binocular tasks such as optical flow estimation, the original CroCo architecture is used as is.

For pre-training, CroCo relies on pairs of images depicting the same scene; in this work the pairs were obtained from synthetic renderings of indoor scenes produced with the Habitat [56] simulator. We empirically show that high masking ratios, *e.g.* 90%, lead to the best pre-training performance. Our model is evaluated on monocular downstream tasks, in particular depth estimation on the NYUv2 dataset [57] and a diverse set of dense 2D and 3D regression tasks taken from Taskonomy [75]. We show that CroCo leads to significantly better performance compared to existing MIM models, whether they were pre-trained on the same data or on ImageNet [55]. When evaluated on monocular high-level semantic tasks, such as ImageNet classification, CroCo obtains lower performance compared to established MIM models, however. This essentially comes from our use of indoor scenes for pre-training, instead of highly semantic datasets like ImageNet, as empirically shown in our experiments. Lastly, we demonstrate that CroCo can be applied to binocular downstream tasks in a straightforward manner without bells and whistles. For instance, optical flow estimation can be performed by directly regressing 2 values per pixel using the decoder, and likewise, relative pose regression is achieved by simply appending a pose regression head to CroCo. In both cases, we show that CroCo pre-training leads to competitive results in these 3D vision downstream tasks.

## 2  Related work

**Self-supervised representation learning** is a paradigm developed to learn visual features from large-scale sets of unlabeled data [36], before finetuning the model on downstream tasks, *e.g.* classification or detection. To achieve that, a *pretext* task is designed and exploits inherent data attributes to automatically generate surrogate labels. The earliest principle behind most existing pretext tasks for computer vision revolves around purposefully removing *some* information from an image, *e.g.* the color, the orientation or the ordering of a sequence of patches obtained from the image that the model then learns to recover [23, 28, 46, 47]. It has been shown that despite the lack of any semantic supervision, networks trained to recover this artificially removed information learn useful representations and facilitate the training of various supervised downstream tasks [1, 26].

Recently, the paradigm of instance discrimination has received a lot of attention, achieving highly competitive results in self-supervised visual representation learning [2, 11, 12, 14, 31, 33]. It seeks to learn outputs that are invariant to well designed classes of data augmentation. Positive image pairs, obtained from the same instance by data augmentation, are pulled closer by the model, while samples obtained from different instances are pushed apart in the embedding space.

More recently, motivated by the success of BERT [19] in NLP and by the introduction of Vision Transformers (ViT) [21], a variety of masked image prediction methods for self-supervised pre-training of vision models has been proposed. Reminiscent of denoising autoencoders [64] or context encoders [49], and aiming to reconstruct masked pixels [3, 13, 21, 25, 32, 70], discrete tokens [7, 79] or deep features [5, 67], these methods have demonstrated the ability to scale to large datasets and models and achieve state-of-the-art results on various downstream tasks. In particular, the masked autoencoder (MAE) [32] accelerates pre-training by using an asymmetric architecture that consists of a large encoder that operates only on unmasked patches followed by a lightweight decoder that reconstructs the masked patches from the latent representation and mask tokens. MultiMAE [4] leverages the efficiency of the MAE approach and extends it to multi-modal and multitask settings. Rather than masking input tokens randomly, the Masked Self-Supervised Transformer model (MST) [38] proposes to rely on the attention maps produced by a teacher network to dynamically mask low response regions of the input, and a student network is then trained to reconstruct it.

**Self-supervised pre-training for dense downstream tasks**. Seminal self-supervised methods [15, 31] were designed to output global image representations, thus encoding limited local information, and thereby hindering transferability of the learned models to downstream tasks involving dense per-pixel predictions, such as semantic segmentation. To alleviate this issue, several self-supervised methods have been proposed to perform contrastive learning on dense local representations rather than global ones [40, 66, 69]. In contrast, InsLoc [72] proposes to paste image instances at various locations and scales onto background images, then predicts instance categories and foreground bounding boxes in the composed images. In VADeR [50] and FlowE [71], dense image representations are learned for several dense semantic downstream tasks such as object detection and semantic segmentation, using multi-hierarchy features [65], while [74] combines multi-resolution parallel modules with local-window self-attention to improve the memory and computation efficiency of dense prediction. These self-supervision methods lead to improved performance on dense semantic downstream tasks such as object detection or semantic/instance segmentation. However, their design and evaluation protocols are not focused on 3D vision and lower-level geometric tasks such as depth, motion and flow estimation.

For such tasks, the self-supervision signal can be obtained via view synthesis, where the model is trained by enforcing photometric consistency [30, 45, 73, 81], or from RGB-D data, where the model is trained to match 2D visual features with 3D geometric representations [24, 34, 41].

## 3 Cross-view Completion Pre-training

In this section we present CroCo, our self-supervised pre-training method based on cross-view completion and tailored to 3D vision tasks.

**Overview**. Figure 3 gives an overview of the CroCo architecture. Let $x_1$ and $x_2$ be two images of the same scene taken from different viewpoints. Both images are divided into $N$ non-overlapping patches $p_1 = \{p_1^1, \ldots, p_1^N\}$, $p_2 = \{p_2^1, \ldots, p_2^N\}$, and a number $n = \lfloor rN \rfloor$ of tokens from the first set $p_1$ is randomly masked, with $r \in [0, 1]$ being a masking ratio hyper-parameter. We typically use $r = 0.9$, $i.e.$, 90% of patches from $p_1$ are discarded. We denote the remaining set of visible patches from the first image by $\tilde{p}_1 = \{p_1^i | m_i = 0\}$, with $m_i = 0$ indicating that the patch $p_1^i$ is not masked ($m_i = 1$ otherwise).

An encoder $\mathcal{E}_\theta$ processes $\tilde{p}_1$ and $p_2$ independently, and a decoder $\mathcal{D}_\phi$ takes the encoding $\mathcal{E}_\theta(\tilde{p}_1)$, conditioned on encoding $\mathcal{E}_\theta(p_2)$, in order to reconstruct $p_1$:

$$\hat{p}_1 = \mathcal{D}_\phi \left( \mathcal{E}_\theta(\tilde{p}_1); \mathcal{E}_\theta(p_2) \right). \tag{1}$$

**Details on the encoder $\mathcal{E}_\theta$**. A Siamese network denoted by $\mathcal{E}_\theta$ with shared weights $\theta$, implemented as a ViT [21], serves to encode the two input images previously split into independent patch sets $\tilde{p}_1$ and $p_2$. In this study, we consider images of resolution $224 \times 224$, with a patch size of $16 \times 16$ pixels. Following ViT, the encoder consists in a linear projection of the flattened input RGB patches,

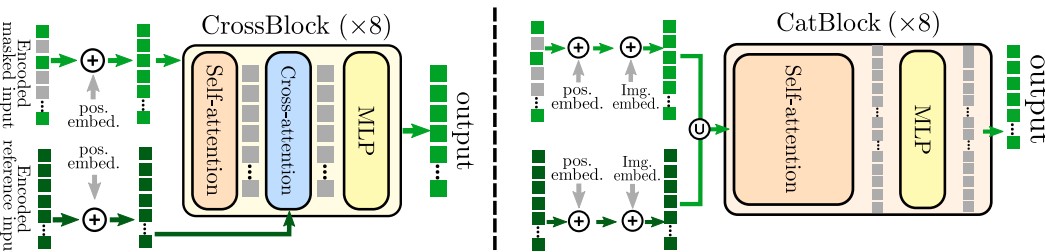

Figure 4: **Attention blocks in the decoder.** CrossBlock (left) combines information from the input sets by alternating self-attention and cross-attention, while CatBlock (right) concatenates the two sets before applying self-attention.

to which sinusoidal positional embeddings are added, followed by a series of transformer blocks [63], each composed of a multi-head self-attention block and an MLP (Multi-Layer Perceptron).

**Details on the decoder $\mathcal{D}_\phi$.** The decoder $\mathcal{D}_\phi$ with weights $\phi$ takes as input the set of encoded tokens from the first image $\mathcal{E}_\theta(\tilde{\boldsymbol{p}_1})$ concatenated with a learned representation $\boldsymbol{e}_{\text{mask}}$ that is repeated $n$ times to account for the masked patches. These tokens are iteratively processed by an attention-based block that also takes into account the encoded tokens $\mathcal{E}_\theta(\boldsymbol{p_2})$ from the reference image $\mathbf{x_2}$. A sinusoidal positional encoding is added to all tokens before being decoded by $\mathcal{D}_\phi$. We experiment with two different architectures for the decoder block that are illustrated in Figure 4, see Appendix B for detailed equations. The first one, termed *CrossBlock*, consists of (a) multi-head self-attention on the tokens representing the first image; (b) multi-head cross-attention between these tokens and the tokens in $\mathcal{E}_\theta(\boldsymbol{p_2})$ from the reference image, and (c) an MLP. The second architecture, termed *CatBlock*, proceeds by concatenating the tokens from the two images, with an added learnable embedding for each of the two images, and then feeding these tokens to a series of standard transformer blocks, *i.e.*, each consisting of (a) multi-head self-attention and (b) an MLP layer. Only the subset of tokens corresponding to the first image is taken into account for the final prediction. These two attention blocks represent a trade-off between model size and computational cost. Indeed, *CrossBlock* has more learnable parameters, due to the cross-attention module, but a lower computational cost as it avoids the quadratic complexity of computing self-attention over the joint token set of size $2N$.

**Training the CroCo model**. The decoder produces one feature vector per patch token, which is processed by a fully-connected layer that outputs 3 values (R, G and B) per pixel for each patch, *i.e.*, $16 \times 16 \times 3 = 768$ values for squared patches with side $16$. These outputs serve as predicted reconstruction $\hat{\boldsymbol{p}_1}$, evaluated using a reconstruction error such as the Mean Squared Error (MSE) loss between predictions and ground-truth values of the pixels, averaged over the set of all masked tokens, denoted $\boldsymbol{p_1}\backslash\tilde{\boldsymbol{p}_1}$. Thus to perform self-supervised pre-training, the network is trained to minimize:

$$\mathcal{L}(\boldsymbol{x_1}, \boldsymbol{x_2}) = \frac{1}{|\boldsymbol{p_1}\backslash\tilde{\boldsymbol{p}_1}|} \sum_{\boldsymbol{p_1^i} \in \boldsymbol{p_1}\backslash\tilde{\boldsymbol{p}_1}} \|\hat{\boldsymbol{p}_1^i} - \boldsymbol{p_1^i}\|^2. \tag{2}$$

We additionally try a variant of the loss where each target patch is normalized using the mean and standard deviation of all pixel values within this patch.

**Pre-training details**. We implement our CroCo model in PyTorch [48] and train the network for 400 epochs using the AdamW optimizer [43]. We use a cosine learning rate schedule with a base learning rate of $1.5 \times 10^{-4}$ for an effective batch size of 256; with a linear warmup in the first 40 epochs. As encoder, we use a ViT-Base/16 backbone, *i.e.*, a series of 12 transformer blocks with 768 dimensions and 12 heads for self-attention with patches of size $16 \times 16$. For both *CrossBlock* and *CatBlock* decoders, we use a series of 8 blocks with 512 dimensions and 16 attention heads. To account for the different dimensions of the encoder and the decoder, we apply a fully-connected layer between them.

**Pre-training data.** We train our model on a set of synthetic image pairs of 3D indoor scenes derived from the HM3D [52], ScanNet [18], Replica [58] and ReplicaCAD [61] datasets as follows. In each 3D scene, we randomly sample up to 1000 pairs of camera viewpoints with a co-visibility greater than 50%, and render these pairs of images using the Habitat simulator [56]. In total, we generated a dataset of 1,821,391 pairs, that we refer to as *Habitat* in this paper.

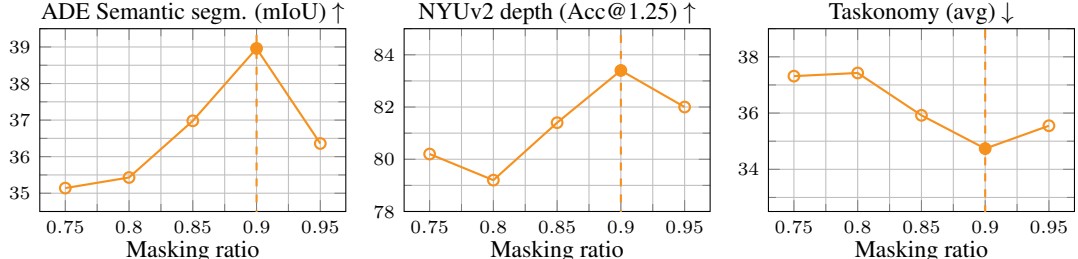

Figure 5: **Impact of the masking ratio** using the CrossBlock decoder and a loss on RGB values.

**Finetuning CroCo.** Our CroCo model can be used as pre-training to solve either monocular or binocular tasks, dense or not. For monocular task, the ViT-Base encoder is used alone, while binocular tasks benefit from the pre-training of both the encoder and the decoder.

## 4 Experimental results

We evaluate CroCo on both monocular downstream tasks in Section 4.1, for which we provide ablations (Section 4.1.1) and a comparison to the state of the art (Section 4.1.2), and on binocular tasks in Section 4.2. Training details for each task are available in the appendix.

### 4.1 Monocular transfer tasks

**High-level semantic tasks**. While targeting 3D vision tasks, we still evaluate some high-level semantic tasks, namely *image classification with linear probing* on ImageNet-1K [55] and report top-1 accuracy. We follow the exact same protocol as MAE [32] for that, with global average pooling for CroCo as we did not include a [CLS] token in our model. Additionally, we report results on *semantic segmentation* on ADE20k [78] with 150 classes and 20,210 training images. We follow the protocol of [4] and report mean Intersection-over-Union (mIoU) on the validation set, using the same ConvNext [42] prediction head on top of the encoder.

**3D vision tasks**. We evaluate *monocular depth prediction* on NYUv2 [57] (795 training and 655 test images) by reporting $\delta_1$, *i.e.*, the percentage of pixels that have an error ratio (max over prediction divided by ground-truth and its inverse) below $1.25$. We additionally report results on the Taskonomy dataset [75] (800 training images) and report L1-loss on the 'tiny' test set (54,514 images) for the same 8 tasks as [4]: *curvature, depth, edges, 2D keypoints, 3D keypoints, normal, occlusion and reshading*. For better clarity with decimal number, we report the L1-losses multiplied by 1000. We also report the average ranking (rank.) over the 8 tasks in tables. For these tasks, we use a DPT head [53] on top of the encoder. We additionally experiment on the monocular task of absolute camera pose regression in Appendix D.4 and the binocular task of stereo image matching in Appendix E.3.

#### 4.1.1 Ablations

**Masking ratio**. We first report in Figure 5 the performance on semantic segmentation (ADE), depth estimation (NYUv2) and Taskonomy when varying the masking ratio $r$ using a *CrossBlock* decoder and a loss without normalization. We observe that the best performance is obtained with a masking ratio of $r = 90\%$. This optimal ratio is higher than what was found for instance in MAE [32] for the auto-completion task. We attribute this to the help provided by the reference image in cross-view completion. We show some reconstruction examples in Figure 2. Despite the small number of visible patches, this amount, combined with the reference view, seems to provide sufficient information to reconstruct the first image. Note that the reconstructions tend to be blurry. This is due to the MSE loss, but as noted in MAE [32], beyond a certain point, sharper reconstructions do not necessarily lead to better pre-training.

**Normalized targets**. Regressing RGB values normalized by the mean and standard deviation within each patch has proven to be effective for MAE [32]. The first two rows of Table 1 compare the results with or without this normalization for CroCo. We observe that it indeed consistently improves the performance on all tasks, and we therefore use it in the remainder of the experiments.

Table 1: **Impact of normalizing targets and decoder block** with a masking ratio of 90%. We also indicate the number of FLOPs and parameters for the full network (encoder and decoder).

| Normalized Target | Decoder Block | ADE ↑ segm. | NYUv2 ↑ depth | Taskonomy ↓ avg. | Taskonomy ↓ rank. | FLOPs | Params |
|---|---|---|---|---|---|---|---|
| | CrossBlock | 39.0 | 83.4 | 34.74 | 2.50 | 50.2G | 120M |
| ✓ | CrossBlock | 40.6 | 85.6 | **33.00** | **1.63** | 50.2G | 120M |
| ✓ | CatBlock | **41.3** | **86.2** | 33.35 | 1.88 | 58.5G | 111M |

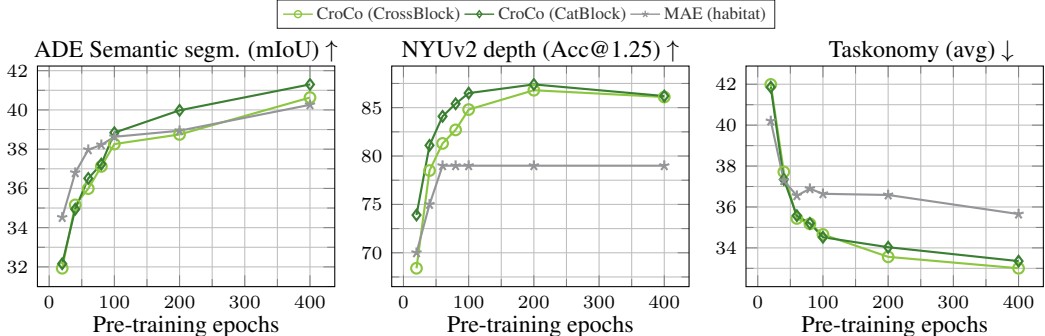

Figure 6: **Performance as a function of the number of pre-training epochs** for various models.

**Decoder architecture**. In Table 1, we compare decoders built with the two proposed attention blocks: CrossBlock which uses cross-attention to exchange information between the two images, and CatBlock which concatenates the tokens from both images. Note that CatBlock has a smaller number of learnable parameters but a higher number of FLOPs. The performances obtained with the two proposed architectures are quite similar; CatBlock yielding slightly better performance for semantic segmentation or depth estimation while CrossBlock performs better on Taskonomy. We favor the use of CrossBlock in what follows. In Appendix C.3, we provide an ablation study on the number of blocks used in the decoder.

**Leveraging the decoder**. Note that the pre-trained decoder can also be used when finetuning the model on monocular tasks: it suffices to feed the input image twice to the whole CroCo network, which in practice is done by duplicating the encoder output before passing it to the decoder. While this significantly increases the computational cost of the model, it yields a consistent gain of performance from 40.6 to 41.0 on ADE, 86.1 to 88.1 on NYUv2, and an improvement on 6 out of 8 tasks on Taskonomy, see further details in Appendix D.2.

**Training profile**. In Figure 6 we show the performance on the downstream tasks as the pre-training progresses up to 400 epochs, when using CrossBlock or CatBlock in the decoder. We observe that the performance on 3D vision tasks – depth estimation on NYUv2 and Taskonomy – reaches a plateau and thus we evaluate our model pre-trained for 400 epochs in what follows. We additionally train a MAE model[1] on the Habitat dataset, using all available images (*i.e.*, twice the number of pairs). When pre-training on Habitat, both models achieve similar performance on the task of semantic segmentation, while CroCo significantly outperforms MAE on depth prediction and Taskonomy. This shows the benefit of using cross-view completion pre-training, rather than pure auto-completion, for 3D vision downstream tasks.

**Training pairs**. CroCo pre-training requires pairs of images, which we obtain in our experiments by sampling two different points of view using the Habitat simulator. As an alternative, we evaluate using geometric transformations (homographies, rotations, scaling, crop, *etc.*) applied to an input image in order to generate the reference view; we report the results in Table 2 after 400 epochs of pre-training on Habitat. We observe a clear drop in performance on all downstream tasks, *i.e.*, semantic segmentation, monocular depth estimation and Taskonomy. One hypothesis is that in this case – when there exists a synthetic transform between the two images – the model can solve the

---

[1]https://github.com/facebookresearch/mae

Table 2: **Comparison with artificial pairs.** We compare pairs obtained by geometric transformations of one image to pairs sampled from two different viewpoints.

| image pairs from | ADE ↑ | NYUv2 ↑ | Taskonomy ↓ | |
|---|---|---|---|---|
| | segm. | depth | avg. | rank. |
| two viewpoints | **38.8** | **86.8** | **33.56** | **1.25** |
| geometric transformations of one image | 27.0 | 66.1 | 47.81 | 1.75 |

Table 3: **Comparison to the state-of-the-art pre-training methods** on semantic tasks (image classification on ImageNet-1K with linear probing and semantic segmentation on ADE) and 3D vision tasks (NYUv2, Taskonomy), with a ViT-Base/16 backbone. We indicate the pre-training data in parenthesis. Best and second best results are **bold** and underlined.

| pre-training method (data) | IN1K ↑ | ADE ↑ | NYUv2 ↑ | Taskonomy ↓ | | | | | | | | | |
|---|---|---|---|---|---|---|---|---|---|---|---|---|---|
| | lin. | segm. | depth | curv. | depth | edges | kpts2d | kpts3d | normal | occl. | reshad. | avg. | rank. |
| DINO [12] (IN1K) | **78.2** | 44.7 | 66.8 | 43.04 | 38.42 | 3.80 | 0.16 | 45.85 | 65.71 | 0.57 | 115.02 | 39.07 | 5.00 |
| MAE [32] (IN1K) | 68.0 | 46.1 | 79.6 | 41.59 | 35.83 | 1.19 | 0.08 | 44.18 | 59.20 | 0.55 | 106.08 | 36.09 | 2.13 |
| MutliMAE [4] (IN1K) | 60.2 | **46.4** | 83.0 | 41.42 | 35.38 | 2.17 | **0.07** | 44.03 | 60.35 | 0.56 | 105.25 | 36.17 | 2.75 |
| MAE (Habitat) | 32.5 | 40.3 | 79.0 | 42.06 | 33.63 | 1.79 | 0.08 | 44.81 | 59.76 | 0.56 | 102.54 | 35.65 | 2.88 |
| **CroCo** (Habitat) | 37.0 | 40.6 | **85.6** | **40.91** | **31.34** | 1.74 | 0.08 | **41.69** | **54.13** | **0.55** | **93.58** | **33.00** | **1.25** |

pretext task of cross-view completion by directly fitting this transform, without reasoning about the geometry of the scene.

This in turn raises the question of how to sample optimal viewpoints when generating synthetic image pairs using the Habitat simulator. To answer it, we study the relation between downstream performance and co-visibility in pre-training image pairs (*i.e.*, how much visual content is shared between the two images) in Appendix C.2. In short, it shows that using image pairs with a co-visibility ratio of approximately 0.5 leads to the best pre-trained models, and that on the one hand, very little co-visibility between the two images is sub-optimal as the task boils down to auto-completion, and on the other hand , if the two images composing a pair overlap too much, performance drops as the pre-training task becomes trivial.

### 4.1.2 Comparison to the state of the art

In Table 3, we compare the proposed CroCo pre-training strategy to the state of the art on monocular tasks. In particular, we compare to DINO [12], a state-of-the-art self-supervised method based on instance discrimination, and to MIM methods with MAE [32] and MultiMAE [4]. The latter uses extra data modalities, namely depth and semantic segmentation generated by off-the-shelf supervised models for pre-training. All three methods rely on the ImageNet-1K (IN1K) dataset [55] for pre-training. This leads to high performance on high-level semantic tasks compared to our approach. Note that DINO, which outputs a global representation, performs best on ImageNet-1K, while MIM-based approaches, which already output dense patch-level predictions during pre-training, obtain better results on dense tasks such as semantic segmentation or depth.

To measure the impact of the pre-training dataset, we report the performance of a MAE model trained on Habitat, *i.e.*, with exactly the same data and ViT-Base architecture as used for CroCo. In this case, the performance on semantic tasks largely drops compared to pre-training on ImageNet (*e.g.* 68.0% to 32.5% in classification on IN1K, see Table 3). This confirms that the high performance on semantic tasks are mostly due to the use of IN1K for pre-training. When evaluating 3D vision downstream tasks, such as depth estimation on NYUv2 or Taskonomy, CroCo significantly outperforms all other methods. Interestingly, it even outperforms MultiMAE for depth estimation on NYUv2 with 85.6% Acc@1.25 *vs*. 83.0%, while other approaches are below 80%. On Taskonomy, CroCo performs best on 6 tasks out of 8 and is second best on the two other tasks, with an average rank of 1.25/5.

### 4.2 Applications to binocular tasks

We now empirically demonstrate that our CroCo model achieves competitive performance in two binocular tasks, namely optical flow and relative pose estimation, without any task-specific design.

Table 4: **Optical flow results** on the training set of the MPI-Sintel dataset for various pre-training methods when finetuned on AutoFlow.

| encoder init. | decoder init. | MPI-Sintel clean | final |
|---|---|---|---|
| random | random | 18.81 | 18.97 |
| MAE (IN1K) | random | 4.68 | 5.16 |
| MAE (Habitat) | random | 4.63 | 5.24 |
| **CroCo** (Habitat) | **CroCo** (Habitat) | **3.00** | **3.60** |

**Optical Flow**. We treat optical flow as a straightforward regression task and do not change the pre-training architecture except for modifying the regression head to output two flow channels instead of 3 RGB color channels. We finetune our network on the public 40,000 synthetic pairs from AutoFlow [59], using a simple MSE loss on $224 \times 224$ crops, for 100 epochs, without any data augmentation besides color jittering. We evaluate on the MPI-Sintel [9] dataset (1041 image pairs in the train split), on the clean and final renderings, using the average endpoint error (AEPE) metric. To test on high resolution $1024 \times 536$ images, we regularly sample overlapping tiles of size $224 \times 224$ at the same position in both input images. We regress the flow between each pair of corresponding tiles. To recombine flow values, at each pixel location in the final prediction we use the flow value predicted by the nearest tile (*i.e.*, based on the tile center).

In Table 4 we compare performance for various pre-training strategies and datasets on the MPI-Sintel training set. Without pre-training, the network is unable to learn anything useful, as shown by the large AEPE values, despite thorough but unsuccessful hyper-parameter tuning. This may be due to the high difficulty of the task coupled with the limited amount of training data. The performance significantly improves when we initialize the encoder with an MAE pre-trained model. With a CroCo initialization of both encoder *and* decoder weights, however, we again observe a large improvement with an average decrease of the error by almost 2 pixels *w.r.t.* the best MAE model on both clean and final renderings. Interestingly, we find that even with just 2 blocks in the decoder, a model pre-trained with CroCo still outperforms an MAE model having 8 decoder blocks (3.59/4.35 AEPE on clean/final, resp., compared to 4.63/5.24 AEPE for MAE; see Appendix E.1 for a complete ablation). These results clearly demonstrate the benefits from pre-training with the cross-view completion task.

Our results (3.00/3.60 on clean/final, resp.) are slightly behind recent state-of-the-art approaches like PWC-Net[60] (2.55/3.93), LiteFlowNet2 [35] (2.24/3.78) and RAFT [62] (1.43/2.71), all trained on FlyingChairs [22] and FlyingThings [44] combined. However, our CroCo model remains competitive for this task considering the simplicity of our approach. In particular, note that (a) our model is trained from limited data without any sophisticated data augmentation, (b) our architecture is generic and not specially designed for optical flow prediction, unlike [60, 35, 62] which all rely on 4D cost volumes, (c) regression at test time is simple and straightforward, (d) the small $224 \times 224$ resolution used at test time hinders our performance on large displacements, a problem which we leave to future work. We refer to Appendix E.1 for additional ablative experiments and comparisons.

**Relative pose estimation**. We experiment on the relative pose regression (RPR) downstream task. Given a pair of images, the goal is to predict the relative pose of the camera with respect to a reference view. To do so, we feed to the CroCo model two images corresponding to normalized camera views, and we regress the rigid transformation $(\boldsymbol{R}, \boldsymbol{t}) \in SO(3) \times \mathbb{R}^3$ between these two views using a differentiable Procrustes layer [8] on top of the prediction head to ensure that $\boldsymbol{R}$ is a rotation matrix. We finetune the RPR model with an MSE loss between the predicted relative pose $(\boldsymbol{R}, \boldsymbol{t})$ and a ground truth $(\hat{\boldsymbol{R}}, \hat{\boldsymbol{t}})$: $\|\boldsymbol{R} - \hat{\boldsymbol{R}}\|_F^2 + \lambda \|\boldsymbol{t} - \hat{\boldsymbol{t}}\|^2$, with $\lambda$ set to 100 in practice. During training, we augment the training image pairs with random permutations, color jittering and virtual camera rotations. We evaluate on the 7-scenes dataset [29], with a single model finetuned on the union of the official training sets of all scenes, and tested on the corresponding test splits. Images are rescaled and cropped to a $224 \times 224$ resolution for simplicity. At test time, for each image of the test split we retrieve the nearest image from the training set according to their AP-GeM-18 descriptors [54], and we predict the pose relative to this retrieved image.

For each scene, we report the standard median position and median orientation error in Table 5. We compare results obtained with the model initialized with CroCo to that of a model with an encoder initialized with MAE (Habitat) and a decoder trained from scratch (last two rows). Unsurprisingly,

Table 5: **Relative pose estimation results** with the median camera position and orientation errors on 7-scenes. Finetuning a model pre-trained with CroCo achieves competitive results compared to existing methods directly regressing relative camera pose, without applying any fusion technique.

| Method / pre-training | chess | fire | heads | office | pumpkin | redkitchen | stairs | Average |
|---|---|---|---|---|---|---|---|---|
| RelocNet* [6] | 12cm, 4.14° | 26cm, 10.44° | 14cm, 10.5° | 18cm, 5.32° | 26cm, 4.17° | 23cm, 5.08° | 28cm, 7.53° | 21cm, 6.74° |
| NC-EssNet* [80] | 12cm, 5.63° | 26cm, 9.64° | 14cm, 10.66° | 20cm, 6.68° | 22cm, 5.72° | 22cm, 6.31° | 31cm, 7.88° | 21cm, 7.50° |
| CamNet*† [20] | 4cm, **1.73°** | **3cm, 1.74°** | 5cm, **1.98°** | 4cm, **1.62°** | **4cm, 1.64°** | **4cm, 1.63°** | **4cm, 1.51°** | **4cm, 1.69°** |
| top1 AP-GeM-18 | 27.9cm, 12.81° | 40.4cm, 16.06° | 21.6cm, 16.46° | 37.5cm, 12.79° | 44.4cm, 12.58° | 46.7cm, 13.92° | 32.2cm, 14.59° | 36cm, 14.2° |
| MAE (Habitat) | 13.2cm, 9.44° | 32.0cm, 15.10° | 16.0cm, 16.75° | 24.8cm, 11.54° | 25.4cm, 10.62° | 29.4cm, 13.32° | 32.8cm, 14.88° | 24.8cm, 13.09° |
| **CroCo** (Habitat) | **2.4cm**, 2.81° | 4.0cm, 3.86° | **3.1cm**, 4.00° | **3.4cm**, 2.53° | 4.9cm, 2.79° | 5.5cm, 3.72° | 11.7cm, 4.53° | 5.0cm, 3.46° |

*: fuse multiple pose predictions     †: exploit temporal information and multi-step retrieval

the former leads to significantly better results than the latter. We furthermore compare our model pre-trained with CroCo to other state-of-the-art RPR methods which directly regress relative camera poses [6, 20] or essential matrices [80] as well as the the retrieval baseline consisting in predicting the identity as relative pose (see Table 5 upper part). Our model outperforms all methods except CamNet [20]. Note that the latter fuses multiple pose predictions, exploit temporal information and multi-step retrieval; in contrast our model directly predicts the relative pose from a (query, map) image pair, without any further processing, and is thus significantly simpler.

## 5 Discussion

We have introduced the novel task of cross-view completion for pre-training computer vision networks tailored to 3D vision downstream tasks. Our experiments show that cross-view completion allows to learn representations better suited to 3D vision tasks than classical MIM with auto-completion, and straightforwardly transfer to monocular and binocular tasks. A limitation of our model pre-trained with cross-view completion is that it seems less tailored to high-level semantic tasks. This is arguably more related to the choice of the dataset than to the pre-training task itself, and future work could explore the use of cross-view completion in conjunction with more object-centric datasets like ImageNet. Our approach requires pairs of images depicting the same scene, and in this work, we leverage synthetic renderings only. Future work could also extend it to real-world image pairs, which can be obtained without any supervision, for instance with Structure-from-Motion [39, 51] or geo-referencing.

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
