# OpenReview forum: "CroCo: Self-Supervised Pre-training for 3D Vision Tasks by Cross-View Completion"
_NeurIPS.cc/2022/Conference — NeurIPS 2022 Accept_

### Official Review · Reviewer_RGgF · 2022-07-10

**Rating:** 6
**Confidence:** 4
**Soundness:** 3 good
**Presentation:** 3 good
**Contribution:** 3 good

**Summary:**

The paper proposes to extend MAE to cross-view completion as a pretext task. The encoder takes in a masked input and unmasked reference view, and the decoder use cross-view attention to reconstruct the input. The paper shows that it improves 3D related monocular tasks comapred to MAE / DINO that do not do cross-view pretraining. The paper also shows comparable results on optical flow, pose estimation, etc.


**Questions:**

See weaknesses 1-3

**Limitations:**

The authors have adequately addressed the limitations and potential negative societal impact of their work.

**Strengths And Weaknesses:**

Strength
+ The paper proposes an interesting idea that extend MAE to mult-view for 3D downstreaming tasks that goes beyonds semantic to more geometric task.
+ The ablations are solid and support their design.
+ The empirical results seem to demonstrate the effectiveness of their method on 3D monocular tasks. e.g. Figure 6, Table 2.

Weakness
1. The method assumes >50% overlaps between the reference view and the target view. I wonder if they have studied how the area affects the performance.
2. The model is trained with a set of synthetic data (specifically indoor scans in this paper), which may limits its application and scalability.
3. I do not fully understand why the masked ratio matters a lot even with cross-attention since the query will have a global “field of view” from the reference image and no direct context from its own. Can the authors comment on that?

---

> ### Author Response · Authors · 2022-08-02
> **Answer to Reviewer RGgF**
>
> ### Overlap between pre-training pairs
>
> > The method assumes >50% overlaps between the reference view and the target view. I wonder if they have studied how the area affects the performance.
>
> We refer to our common answer to all reviewers.
>
> ### Pre-training on synthetic pairs
>
> > The model is trained with a set of synthetic data (specifically indoor scans in this paper), which may limits its application and scalability.
>
> Similarly, please refer to our common answer.
>
> ### Masking ratio
>
> > *I do not fully understand why the masked ratio matters a lot even with cross-attention since the query will have a global “field of view” from the reference image and no direct context from its own. Can the authors comment on that?*
>
> The masking ratio is useful to balance two opposing principles of the cross-view completion pre-training.
>
> On the one hand, the cross-view completion problem is trivial when most patches of the first view are visible. Indeed, the model could simply ignore the reference view and return the input image (with some minor inpainting). Thus, masking most patches from the first view limits the amount of provided context and encourages the model to perform cross-view reasoning to decrease the reconstruction cost, which requires understanding geometric and semantic cues, leading to better (in the sense of being more transferable) representations.
>
> On the other hand, the cross-view completion problem cannot be solved either if the first image is fully masked. In such case, the model has no way to correctly predict the viewpoint change *w.r.t.* the reference view is necessary to reconstruct the target image. CroCo thus requires at least some patches of the first image to remain visible. These visible patches should provide enough context regarding the target view to estimate the changes of viewpoint and illumination *w.r.t.* the reference view. Visible patches can additionally provide some useful context to reconstruct parts of the masked image that are non-observable in the reference view, since the reference view does not necessarily have a larger field-of-view than the masked view (i.e., parts of the masked view can be occluded or out-of-frame in the reference view).
>
> Both of these extremes (no mask or full mask) lead to degenerate settings, but suggest the existence of a *sweet spot* in terms of masking ratio somewhere in-between, which is confirmed empirically.

---

### Official Review · Reviewer_nsnJ · 2022-07-11

**Rating:** 6
**Confidence:** 4
**Soundness:** 3 good
**Presentation:** 3 good
**Contribution:** 3 good

**Summary:**

This paper presents CroCo, a new self-supervised learning approach for pretraining transformers for low-level and 3D computer vision tasks. The design of the approach is similar to the recent Masked Autoencoder (MAE), but uses two images corresponding to two views of the same scene (with some overlap) rather than single images. The main novelty is conditioning the prediction of the masked patches of an image on another complete view of the scene, in addition to the visible patches of the image and their spatial encodings. This allows the model to use the information in this second view to predict the masked patches, if it can successfully reason about the spatial relationships between the two views.

The proposed method is evaluated on an array of tasks, including standard ImageNet classification, depth estimation on NYUv2, 8 geometric tasks from Taskonomy, and camera pose estimation. On the tasks that require geometric reasoning, CroCo consistently outperforms MAE.

**Questions:**

1. What are ways in which the limitation of requiring a simulator like Habitat (which in itself is "expensive" as it requires labor intensive scans of indoor 3D scenes) can be addressed? How does this impact the downstream applications that this model is suitable for?

**Limitations:**

The authors have discussed some limitations of their work throughout the draft. The paper itself doesn't have high potential for negative societal impact.

**Strengths And Weaknesses:**

## Originality

### Strengths
This paper follows a long stream of self-supervised learning works targeting image classification and dense prediction tasks like detection and segmentation. This work builds on the MAE idea of predicting masked patches of an image, but conditions this prediction on additional view of the scene. This is very good idea for self-supervised pre-training of representations for downstream tasks that involve reasoning about the geometric properties of images, because it makes sense that in order to use the information in this additional view, the representation has to encode information about the spatial/geometric properties of each view.

### Weaknesses
There is some related work that is highly relevant but not properly cited. These are recent works that tackle self-supervised learning from RGBD data. The authors do not need to compare with these methods, but it would be beneficial to include them in the related work.

El Banani, Mohamed, and Justin Johnson. "Bootstrap your own correspondences." Proceedings of the IEEE/CVF International Conference on Computer Vision. 2021.

Liu, Yunze, et al. "P4contrast: Contrastive learning with pairs of point-pixel pairs for rgb-d scene understanding." arXiv preprint arXiv:2012.13089 (2020).

Hou, Ji, et al. "Pri3d: Can 3d priors help 2d representation learning?." Proceedings of the IEEE/CVF International Conference on Computer Vision. 2021.

## Quality

### Strengths
1. _Motivation and technical implementation of main idea_: As mentioned in the originality section, condition on a complete 2nd view for prediction of masked patches is well motivated. The paper proposes two different decoder modules to combine the information from the masked and 2nd view, CrossBlock and CatBlock, and empirically evaluates the performance/computational cost trade offs between each module (Table 1). The paper presents high quality ablations of masking ratio and target normalization, as well as showing downstream performance as a function of amount of pretraining, which ensures that baseline methods have been trained enough.
2. _Evaluation_: The datasets used for evaluation as well as the baselines are well chosen, with comparisons being made with MAE, DINO, MultiMAE trained on ImageNet and MAE trained on Habitat (Habitat is the main training set for CroCo). Using both the semantic and geometric tasks confirms that masked patch prediction conditioned on an additional scene view is a significantly better choice for pretraining geometric tasks than ImageNet.
3. _Performance_: The margin of performance improvement over the standard MAE is significant

### Weaknesses
1. _Comparison relative to fully supervised methods_: While the authors provide comparisons with supervised baselines for camera pose estimation, it would be highly beneficial to include such results to serve as a reference for the other tasks as well.
2. _Current reliance simulators like Habitat to generate multiple views_: The current training setup requires access to a scene data simulator like Habitat to generate pairs of overlapping views. This potentially limits the applicability of CroCo to more complex, non-indoor scene domains.

## Clarity

### Strengths
The paper is very well written and easy to understand. The architecture diagrams, visualizations and supplementary video are high quality and help with understanding the paper. The plots and tables are clear.

Minor weakness: It might be useful to label the rows of Figure 2 on the figure itself rather than the caption.

## Significance

### Strengths
This work clearly demonstrates the utility of using an additional scene view for self-supervised training via masked patch prediction. Further, the empirical analysis shows that existing masked  patch prediction methods are less suitable for spatial/geometric tasks.

---

> ### Author Response · Authors · 2022-08-02
> **Answer to Reviewer nsnJ**
>
> ### References on self-supervised learning from RGBD data
>
> Thanks for the suggestion, we have added  these references in the related work section of the revised paper (see the updated pdf).
>
> ### Comparison to the state of the art
>
> > *Comparison relative to fully supervised methods: While the authors provide comparisons with supervised baselines for camera pose estimation, it would be highly beneficial to include such results to serve as a reference for the other tasks as well.*
>
> We believe that the reviewer means that Table 4 also provides a comparison to methods dedicated to the task (with task-specific designs), and not just ablations on the pre-training objective.
> * For NYUv2 depth estimation, we report 85.6 in Table 2 and even reach 88.1 in Table B of the supplementary material. The concurrent work of [a] is state of the art on this task with 91.1 Acc\@1.25; this is in part attributed to the use of a MiT-B4 backbone from [b] which has almost three times the number of FLOPs compared to the ViT-Base backbone we use (i.e. 95.7 GFLOPs versus 35.3 GFLOPs).
> * For taskonomy, state-of-the-art results rely on a different setup with more training images and using a multi-task objectives. On our side we followed the setup of MultiMAE (training independently each task on taskonomy-tiny) for evaluating pre-training and we are not aware of any other work using the same setup.
> * For optical flow estimation, other methods are trained using a different setup which includes the MPI-Sintel training dataset, that we use for evaluation in our case. We plan to experiment with another setup in the future.
> * In Table H of the supplementary material, we provide comparative results for stereo matching and also report results for the state of the art.
>
> [a]: Depthformer: Multiscale Vision Transformer For Monocular Depth Estimation With Local Global Information Fusion, Agarwal and Arora, arXiv'22.
> [b]: SegFormer: Simple and Efficient Design ForSemantic Segmentation With Transformers, Xie et al., NeurIPS'21.
>
> ### Synthetic pre-training data
>
> > The current training setup requires access to a scene data simulator like Habitat to generate pairs of overlapping views. This potentially limits the applicability of CroCo to more complex, non-indoor scene domains.
>
> We refer to our common answer to reviewers.
>
> ### Simulator requirement
>
> > What are ways in which the limitation of requiring a simulator like Habitat (which in itself is "expensive" as it requires labor intensive scans of indoor 3D scenes) can be addressed? How does this impact the downstream applications that this model is suitable for?
>
> We believe that the fact that the synthetic data comes from intensive scans of indoor scenes is not critical and expect that generating synthetic pairs from purely virtual world should perform similarly.
>
> As mentioned in our common answer, we are now focusing on generating a large number of real image pairs to actually get rid of the simulator in future work, e.g. leveraging structure-from-motion or unsupervised image matching techniques to generate adequately overlapping pairs from real-world datasets.

---

> > ### Comment · Reviewer_nsnJ · 2022-08-08
> > **Response to rebuttal answer**
> >
> > Thank you for your detailed response. I am overall satisfied with the answers to my questions. In the next version of the paper (or the supplement) please include an explicit discussion about the performance of CroCo w.r.t. supervised baselines for the tasks where this is applicable, as you have done here in the rebuttal.
> >
> > I take some issue with the claim that a purely virtual world would suffice to achieve similar performance on real world downstream tasks, especially due to domain shift issues at the low/image patch level (which is exactly why Habitat is a good choice and probably transfers well). However, training CroCo with extremely photorealistic (but still fully virtual) simulators like HyperSim (https://github.com/apple/ml-hypersim) might lead to similar performance. Is there a specific reason why we should expect CroCo to transfer well when not trained on highly photorealistic data?
> >
> > It’s encouraging to hear that the work is being extended to 100% real data. Thanks again!

---

> > > ### Author Response · Authors · 2022-08-09
> > > **About domain gaps in virtual data**
> > >
> > >
> > > Thank you for your answer.
> > >
> > > We agree that the domain shift can be greater with a fully virtual simulation than with data rendered from real world 3D scans or extremely photorealistic settings. Still, we believe that, if the most interesting part of the signal (the geometry) is present in a given setup (fully virtual, rendered or real world images), the approach can still work, with a moderate drop in performance.
> > > In the paper we show that using data rendered from 3D scans already transfers well despite possible domain gaps.
> > > We will definitely compare to real data, and and possibly to virtual renderings of different quality, in the future,  as indeed could be an interesting experiment to evaluate how would the domain gap affect the downstream tasks.
> > >
> > > Thanks again for raising this point, and for the rest of the feedback given.

---

### Official Review · Reviewer_BHD4 · 2022-07-12

**Rating:** 6
**Confidence:** 5
**Soundness:** 4 excellent
**Presentation:** 3 good
**Contribution:** 3 good

**Summary:**

This paper presents a novel unsupervised representation learning task that utilizes pairs of images showing the same scene from different viewpoints, unlike the existing solution that uses a single image itself for pre-training. The proposed cross-view completion pretext task helps the networks to focus on other image, as well as visible content of input, which in turn improve the pre-training ability. Experiments shows that the proposed pretext task has shown the superiority performance compared to existing methods.

**Questions:**

- For the pre-training dataset, the authors used "synthetic" image pairs of 3D indoor scenes driven from 3D models, such as ScanNet or Replica, which makes sense. But, we also has many "real" image pairs of 3D scenes, such DTU [CVPR'14]. It would be interesting if the performance with such real cross-view dataset is evaluated.
- Similarly to above, idea of using "another" images for Masked Image Modeling (MIM)-like pretext task is very interesting, which definitely reduce the ambiguity of existing MIM-like pre-training. Then we may consider other interesting setting in a manner that we may use a synthetic warping to generate "synthetic" cross-view image, similarly to geometric matching [rocco, CVPR'17]. Such a framework can also be applied to ImageNet benchmark as well. In the current version, Habitat dataset is only used for pre-training for most experiments. It would be very interesting if additional comments or experiments are done regarding this.
- The constraint of "co-visibility greater than 50%" seems heuristic. The additional experiments as varying the such co-visibility ratio, e.g., 10% or 90%, would be very interesting.
- One more interesting thing is that in MAE [CVPR'22], masking ratio of 75% was an optimal in some cases. But in this paper, 90% masking is considered as optimal, mainly based on the analysis in Fig. 5. Is there any dataset dependency of decoder dependency for this? It would be very interesting and important experiments and analysis.
- What happens if we just use a pure decoder? or decoder from MAE [CVPR'22]?
- Computational complexity would be compared.

Minor comments:
- Masking schedule will be also very interesting issue, e.g., dynamic masking ratio?
- Cross-view vs. Multi-view? clarify this difference. In my opinion, "multi-view" is also fine.

**Ethics Review Area:**

["I don’t know"]

**Limitations:**

The authors mentioned the limitations of this paper.

**Strengths And Weaknesses:**

+ Using the cross-view completion as a novel pretext task is very interesting and makes sense.
+ The state-of-the-art performance was attained, even though the methodology is very simple (it can be easily implemented similarly to MIM).
+ Two novel attention blocks in the decoder, which are tailored to cross-completion tasks, are interesting and makes.
+ The paper is well written and easy to follow.

---

> ### Author Response · Authors · 2022-08-02
> **Answer to Reviewer BHD4 (2/2)**
>
> ### Pure decoder architecture
>
> > *What happens if we just use a pure decoder?*
>
> In CroCo, we propose an architecture with an encoder-decoder structure, i.e., where the two images are first encoded independently, before they interact in the decoder. It would indeed be possible to use an architecture where tokens from both images are processed jointly from the beginning of the network and we are planning to try it in the future. However, using an encoder-decoder structure allows a straightforward use of the mode for monocular tasks, which would be less obvious with a pure decoder architecture. Additionally, using a standard vision architecture (ViT) in the encoder enabled us to fairly compare to state-of-the-art pre-training strategies such as MAE.
>
> > *or decoder from MAE [CVPR'22]?*
>
> Note that the so-called *decoder* from MAE is a stack of standard transformer blocks without cross-attention (i.e. a Transformer Encoder), and not a decoder in the sense used in the NLP community per-say: it only process tokens from one single image. The decoder of our CroCo model has similar hyperparameters as MAE's decoder (8 blocks, 512 channels, 16 heads). The main difference is that we change the decoder blocks to either CrossBlocks (which also include a cross-attention mechanism) or CatBlocks (where tokens of the second image are concatenated with the one of the first image, but the blocks remain the same). In the case of CatBlocks, the architecture of the decoder is actually the same as the decoder from MAE, the only difference being the input to this network, i.e., concatenation of the tokens from the first and second images, instead of simply tokens of a given image.
>
>
> ### Computational complexity
>
> > Computational complexity would be compared.
>
> In Table 1 we provide numbers for FLOPs and parameters for the different encoder-decoder architectures considered in our experiments (see details in the supplementary material). For monocular downstream tasks, the complexity of our pretrained model is reduced to the complexity of its encoder: a ViT-Base/16 architecture in our case. Comparisons with other pretraining methods (in Tables 2, 3 and with MAE in Table 4) are thus made with methods of similar test time complexity.
>
> ### Masking schedule
>
> > Masking schedule will be also very interesting issue, e.g., dynamic masking ratio?
>
> This is a great suggestion. We did not study these aspects due to limited computational budget, but it would indeed be interesting to explore.
>
> ### Cross-view vs. Multi-view
>
> > Cross-view vs. Multi-view? clarify this difference. In my opinion, "multi-view" is also fine.
>
> We chose the term *cross-view* because a) only two views are considered, and b) the proposed approach is asymmetrical: one view is used to complete the other. We feel that the term *multi-view* is somewhat more generic, as it can mean more than two views, and also that 'cross' better conveys the notion of asymmetry, as it is typically used in a context that involves an origin and a target (e.g 'crossing the street').

---

> ### Author Response · Authors · 2022-08-02
> **Answer to Reviewer BHD4 (1/2)**
>
> ### Synthetic pre-training pairs
>
> > For the pre-training dataset, the authors used "synthetic" image pairs of 3D indoor scenes driven from 3D models,... It would be interesting if the performance with such real cross-view dataset is evaluated
>
> We refer to our common answer to reviewers.
>
> ### Warping to generate artificial real pairs
>
> >we may consider other interesting setting in a manner that we may use a synthetic warping to generate "synthetic" cross-view image, similarly to geometric matching [rocco, CVPR'17]. Such a framework can also be applied to ImageNet benchmark as well. In the current version, Habitat dataset is only used for pre-training for most experiments. It would be very interesting if additional comments or experiments are done regarding this.
>
> Generating synthetic pairs from real (single) images (e.g. from ImageNet) by applying random homographies is indeed a relevant suggestion, as it may allow to easily generate artificial training data. Unfortunately, our experiments have shown that a model pretrained on such data performs poorly. Namely, in our experiments, pre-training for 200 epochs using homographies leads to around 26% of mIoU on ADE semantic segmentation (compared to around 39% of mIoU with our training pairs) and an average loss of 48 for taskonomy (compared to 34 with our training pairs).
>
> Our hypothesis is that deducing the parameters of the homography from the input pair is easy for the network, which can use it to straightforwardly solve the pretext task almost perfectly. Note that homographies correspond to a change of viewpoint when assuming a planary scene, which is far simpler than with realistic 3D scenes. Consequently homography-based pairs require no semantic nor geometric priors and lead to poor performance on 3D downstream tasks. In summary, training on image pairs featuring complex 3D layouts (as reported in the paper) leads to substantially better performance. This is an interesting ablation, hence we will include these results in the supplementary material of the final version if accepted.
>
>
> ### Overlap between the pre-training image pairs
>
> > The constraint of "co-visibility greater than 50%" seems heuristic. The additional experiments as varying the such co-visibility ratio, e.g., 10% or 90%, would be very interesting.
>
> We refer to our common answer to reviewers.
>
> ###  Masking ratio
>
> > One more interesting thing is that in MAE [CVPR'22], masking ratio of 75% was an optimal in some cases. But in this paper, 90% masking is considered as optimal, mainly based on the analysis in Fig. 5. Is there any dataset dependency of decoder dependency for this? It would be very interesting and important experiments and analysis.
>
> The additional information coming from the reference view is definitely a part of why CroCo supports a higher masking ratio than MAE. Having further redundancy in the input makes the task easier, which is compensated by increasing the masking ratio. This is in line with recent findings in concurrent works on MAE for videos [a,b,c] which also use a masking ratio of 90% to 95%.
>
> In principle the optimal masking ratio should  depend both on the choice of dataset, and on the architecture. Still, in practice we expect similar values to work well for most datasets, but we will certainly ablate this when working with other datasets in future, and in particular in the case of real-world data.
>
> [a]: VideoMAE: Masked Autoencoders are Data-Efficient Learners for Self-Supervised Video Pre-Training, Tong et al., arXiv 2022.
> [b]: Masked Autoencoders As Spatiotemporal Learners, Feichtenhofer et al., arXiv 2022.
> [c]: MaskViT: Masked Visual Pre-Training for Video Prediction, Gupta et al., arXiv 2022.

---

### Official Review · Reviewer_2MEW · 2022-07-17

**Rating:** 5
**Confidence:** 4
**Soundness:** 2 fair
**Presentation:** 3 good
**Contribution:** 3 good

**Summary:**

The article proposes a new model for pretraining 3D vision downstream tasks with Masked Image Modelling. The authors focus on improving the performance for monocular 3D vision downstream tasks such as depth estimation.  Even though the article has appropriately presented the intended ideas, the following comments can be considered to improve the article.

**Questions:**

Introduction
1.	Line 40, the word “instead” could be removed as it does not add meaning to the introduction of the proposed model
2.	Alignment of Figure 2 and 3 could be changed in a closer view with the in-text citation and the corresponding abbreviation of “Croco” could be given before describing both figure 2 and 3 as this term is used in image labelling’s.
3.	Line 52 , as per the understanding and explanation given about  the proposed model, it “first(masked) image” to be mentioned instead of “first(target) image”
Related works
1.	Outcome (research gaps) of related works or summary of related works can be added at the end of the related works section.
Cross-view Completion pretraining
1.	In subsection, Details on the decoder in line 144, more detailed explanation on the self-attention and multi-head attention block working could be given. As they contribute majorly in prediction of the masked area on the image.
2.	In line 154, the CatBlock proceeds to multi-head self-attention and MLP layer but it is not clearly shown in Figure 4
3.	What is the difference between multi-head cross-attention and multi-head self-attention. It would be better if more explanation is given


**Ethics Review Area:**

["I don’t know"]

**Limitations:**

It will be good if the difference between downstream tasks and downstream classifiers be mentioned in self-supervised setting.

**Strengths And Weaknesses:**

Abstract
1.	It is suggested to include the highlight of the results obtained using the proposed model. And, more detailed explanation of the constraints of the existing methods could be reduced as it is repeated in introduction as well


1.	In table 1, the last two rows show the impact of this normalization on the Croco model pretraining. But in line 217, it is mentioned as first two rows
2.	Figure 5 and 6 could have the Y -axis label mentioned on the graphs to get a clear visibility


1.	List of acronyms can be displayed in the appendix.
2.	Punctuation operators like comma, spacing in equations, math notations, and between lines should be thoroughly checked for correctness.

---

> ### Author Response · Authors · 2022-08-02
> **Answer to Reviewer 2MEW**
>
> > *Abstract. It is suggested to include the highlight of the results obtained using the proposed model.*
>
> This was the intent of the last two sentences of our abstract; we will work on improving them accordingly.
>
> > *And, more detailed explanation of the constraints of the existing methods could be reduced as it is repeated in introduction as well*
>
> Stating the limitations of previous work is a simple way to motivate the use of pairs and thus we felt that it needs to be mentioned briefly in the abstract and with more detail in the introduction, but we are open to a different phrasing that would avoid this repetition.
>
> > *In table 1, the last two rows show the impact of this normalization on the Croco model pretraining. But in line 217, it is mentioned as first two rows*
>
> The sentence has been rephrased in the revised paper to improve clarity. We meant that the first two rows correspond to the ablation from which the impact of the normalization can be measured. The last two rows indeed have normalization but ablate the decoder block.
>
> > *Figure 5 and 6 could have the Y -axis label mentioned on the graphs to get a clear visibility*
>
> The metrics reported along the y axis are shown in the title of the plots to maximize the size of the figure. Putting them vertically along the y axis is possible but will lead to slightly reduced size of the plots and vertical texts are sometimes not easy to read. We are open to make the change it in a revised version if necessary.
>
> > *List of acronyms can be displayed in the appendix.*
>
> We will add a table in the appendix with the list of acronyms regularly used in the paper (MIM for Masked Image Modeling, MAE for Masked Auto-Encoder, ViT for Vision Transformers, and CroCo for Cross-view Completion mainly).
>
> > *Punctuation operators like comma, spacing in equations, math notations, and between lines should be thoroughly checked for correctness.*
>
> Thanks, we sought to correct these in the revised paper.
>
> > *Line 40, the word “instead” could be removed as it does not add meaning to the introduction of the proposed model*
>
> Thanks for the suggestion, this is done in the revised paper.
>
> > *Alignment of Figure 2 and 3 could be changed in a closer view with the in-text citation and the corresponding abbreviation of “Croco” could be given before describing both figure 2 and 3 as this term is used in image labelling’s.*
>
> Thanks for the notification; both Figure 2 and Figure 3 have been moved one page up, and the CroCo abbreviation is now introduced before referencing Figures 2 and 3.
>
> > *Line 52, as per the understanding and explanation given about the proposed model, it “first(masked) image” to be mentioned instead of “first(target) image”*
>
> We included a new phrasing without the word *target* in the revised paper to improve clarity, thanks. We meant that the first image, which is masked, also serves as target for the reconstruction.
>
> > *Outcome (research gaps) of related works or summary of related works can be added at the end of the related works section.*
>
> This is a great suggestion; still we had to skip it due to lack of space and believe that the differences between related work and the proposed method remains clear enough.
>
> > *In subsection, Details on the decoder in line 144, more detailed explanation on the self-attention and multi-head attention block working could be given. As they contribute majorly in prediction of the masked area on the image. In line 154, the CatBlock proceeds to multi-head self-attention and MLP layer but it is not clearly shown in Figure 4. What is the difference between multi-head cross-attention and multi-head self-attention. It would be better if more explanation is given*
>
> Multi-head cross- and self-attention refer to the attention mechanism described in the seminal work of Vaswani et al. [57]. Self-attention refers to an attention mechanism applied within one set of tokens, i.e., queries, keys, values all come from the same set of tokens. Cross-attention refers to the fact that two different sets of tokens are involved, a first set for queries (corresponding to tokens of the masked view in CrossBlock) and a second set for keys and values (corresponding to tokens of the reference view in CrossBlock). We will try to improve Figure 4 to make it clearer and will add detailed equations in the supplementary material.
>
> > *It will be good if the difference between downstream tasks and downstream classifiers be mentioned in self-supervised setting.*
>
> Our method uses a self-supervised pre-training setting which can be beneficial for different downstream tasks. These downstream tasks include, but are not limited to, classification. We are not certain to have completely understood the question; we hope we answered and clarified it.  If not, please do not hesitate to rephrase the question so that we can answer during the discussion phase.

---

### Author Response · Authors · 2022-08-02
**Common answer to reviewers**

We thank all the reviewers for their positive feedback and valuable suggestions. We have taken into account comments relating to typos and phrasing in a revised version of the paper, with changes highlighted in blue.

Before providing individual answers to each review, we address concerns shared by multiple reviewers, about 1) the fact that we use synthetic data for generating pre-training pairs and 2) the choice of the overlap ratio between the viewpoints of these training pairs.

### Using synthetic pre-training data

In contrast to existing MIM-based approaches, our proposed cross-view completion (CroCo) pretext task uses for pre-training synthetic image pairs that have the advantage of being easier to acquire/generate.
One obvious drawback is that this introduces an undesirable domain gap between these synthetic images and the real data used in the downstream tasks.
Yet, our CroCo pre-training yields strong results on most 3D downstream tasks, and using synthetic pairs thus offers an interesting compromise between performance and ease of acquisition.
We believe this highlights the promising potential of our approach.

Having said that, it is clear that using real data for pre-training is an interesting direction to further improve performance. Training on real pairs poses the non-trivial problem of how to acquire the appropriate data at scale, which we are currently investigating. One potential direction is to use for instance structure-from-motion (SfM) techniques, as it was done for building landmarks retrieval datasets such as SfM-120k [a] or 3D reconstruction datasets such as MegaDepth [b]. Note that SfM techniques, for instance Colmap, are unsupervised as they rely on handcrafted features such as SIFT [c], so building pairs this way does not require manual labeling. We are also investigating the use of modern unsupervised matching techniques.

Using SfM-based techniques, we have already obtained about a few hundred thousands pairs from the ARKitSCenes dataset [d] and added them to our pre-training data. We observe better performances with for instance +2% mIoU on semantic segmentation on ADE20k and +2% Acc @ 1.25 on NYUv2 monocular depth estimation. We plan to further study the use of real data in the future.

[a]: Fine-tuning CNN Image Retrieval with No Human Annotation, Radenovic et al, IEEE trans. PAMI 2018.
[b]: MegaDepth: Learning Single-View Depth Prediction from Internet Photos, Li and Snavely, CVPR'18.
[c]: Sift-the scale invariant feature transform, Lowe, IJCV'04.
[d]: ARKitScenes - A Diverse Real-World Dataset For 3D Indoor Scene Understanding Using Mobile RGB-D Data, Baruch et al., NeurIPS'21.

### Overlap between the pre-training image pairs

Reviewers have noted that the minimum overlap (i.e. co-visibility) constraint of 50% between the synthetic pre-training pairs appears heuristically chosen, and that more ablations would be beneficial.

We agree that a discussion about the control of the overlap ratio would be beneficial; it was guided by two important observations:
1. if the two images composing a pair overlap too little, the task boils down to auto-completion, therefore we set a threshold on the minimal co-visibility ratio (50%).
2. if the two images composing a pair overlap too much, the task becomes trivial, therefore we encourage large viewpoint changes between images in our training set.

We agree that quantifying the ideal distribution of "good pairs" more precisely would be valuable. We will thus run additional ablations for the final version.

---

### Meta-Review · Area_Chair_T1Ad · 2022-08-28

**Recommendation:** Accept
**Confidence:** Certain

**Metareview:**

Combining masked image modeling with cross-view completion, the paper develops a self-supervised pretext task appropriate for learning visual representations for downstream 3D tasks.  After the author response and discussion period, all four reviewers give positive ratings.  The pretext task design is novel, well-motivated for 3D representation learning, and shown to be experimentally effective on downstream 3D vision tasks.

**Award:**

No

---

### Decision · Program_Chairs · 2022-09-14

Accept